



# Variable effects of spatial resolution on modeling of nitrogen oxides

Chi Li[1], Randall V. Martin[1], Ronald C. Cohen[2,3], Liam Bindle[1], Dandan Zhang[1], Deepangsu Chatterjee[1], Hongjian Weng[4], and Jintai Lin[4]

[1]Department of Energy, Environmental  Chemical Engineering, Washington University in St. Louis, St. Louis, MO, USA
[2]Department of Chemistry, University of California, Berkeley, Berkeley, CA, USA
[3]Department of Earth and Planetary Science, University of California, Berkeley, Berkeley, CA, USA
[4]Laboratory for Climate and Ocean-Atmosphere Studies, Department of Atmospheric and Oceanic Sciences, School of Physics, Peking University, Beijing, China

**Correspondence:** Chi Li (lynchlee90@gmail.com)

**Abstract.** The lifetime and concentration of nitrogen oxides ($NO_x$) are susceptible to non-linear production and loss, and consequently to the resolution of a chemical transport model (CTM). Here we use the GEOS-Chem CTM in its high performance implementation (GCHP) to investigate $NO_x$ simulations over the eastern United States across a wide range of resolutions (13-181 km). Following increasing grid size, daytime surface $NO_x$ concentrations over July 2015 generally decrease over the Great

Lakes (GL) region and increase over the Southern States (SS), yielding regional biases (181 km vs. 13 km) of -18% to 9%; meanwhile hydrogen oxide radicals ($HO_x$) increase over both regions, consistent with their different chemical regimes. Nighttime titration of ozone by surface nitric oxide (NO) was found to be more efficient at coarser resolutions, leading to longer $NO_x$ lifetimes and higher surface concentrations of nitrogen dioxide ($NO_2$) over the GL in January 2015. The tropospheric $NO_2$ column density at typical afternoon satellite overpass time has spatially more coherent negative biases (e.g., -10% over

the GL) at coarser resolutions in July, which reversed the positive biases of surface $NO_x$ over the SS. The reduced $NO_x$ aloft (> 1km altitude) at coarser resolutions was attributable to the enhanced $HO_x$ that intrudes into the upper troposphere. Application of coarse resolution simulations for interpreting satellite $NO_2$ columns will generally underestimate surface $NO_2$ over the GL and overestimate surface $NO_2$ over the SS in summer, while uniformly overestimating $NO_x$ emissions over both regions. This study significantly broadens understanding of factors contributing to $NO_x$ resolution effects, and the role of fine resolution to

accurately simulate and interpret $NO_x$ and its relevance to air quality.

## 1    Introduction

Nitrogen oxides ($NO_x \equiv NO + NO_2$) have major roles in tropospheric chemistry and air quality. During daytime, $NO_x$ interacts with hydrogen oxide radicals ($HO_x \equiv OH + HO_2$) and volatile organic compounds (VOCs) via photochemical reactions to affect formation of ozone and nitrate aerosols (e.g., Sillman, 1999; Thornton et al., 2002; Pusede et al., 2015; Zhu et al.,

2022). During nighttime, persistent $NO_x$ emissions are the main chemical sink of ozone in urban areas (Zhang et al., 2004; Brown et al., 2006; Zakoura and Pandis, 2018), meanwhile the sequential formation of nitrate radical ($NO_3$) significantly alters nocturnal atmospheric oxidation capacity and secondary aerosols (Evans and Jacob, 2005; Brown and Stutz, 2012; Rollins et al., 2012). $NO_x$ has strongly localized emissions (Miyazaki et al., 2017; Crippa et al., 2018; Beirle et al., 2019) and



relatively short lifetimes (Kenagy et al., 2018; Laughner and Cohen, 2019), which determine its strong spatial heterogeneity.

While such localization is advantageous for identifying and quantifying $NO_x$ emissions using observations such as satellite nitrogen dioxide ($NO_2$) column density (e.g., Martin et al., 2006; Cooper et al., 2017; Goldberg et al., 2019; Laughner and Cohen, 2019; Wang et al., 2022), it poses as a challenge for chemical transport models (CTMs) to accurately represent the relevant production and loss processes at inter-urban scales (order 10 km) due to limited computational resources.

One outstanding issue is the resolution dependence of simulated $NO_x$ lifetime ($\tau$) (Charlton-Perez et al., 2009; Valin et al.,

2011), which is sensitive to $NO_x$ abundance itself due to the interaction of $NO_x$ with its own chemical loss (Laughner and Cohen, 2019; Shah et al., 2020) such as hydroxyl radical (OH) and ozone (e.g., Figure S1). Due to the stronger $NO_x$ localization at higher resolution, systematic differences in the simulated $\tau$ and $NO_x$ concentration at different resolutions were reported from modeled $NO_x$ plumes of power plants, cities and ship emissions (Sillman et al., 1990; Charlton-Perez et al., 2009; Valin et al., 2011). Regional chemical transport modeling with more realistically distributed emissions was also performed to

examine how $NO_x$ abundance changes with varying resolutions (Valin et al., 2011; Yamaji et al., 2014; Yan et al., 2016; Yu et al., 2016). The majority of these studies indicated increased $\tau$ and $NO_x$ concentration at higher resolutions, and attributed it to the titration of OH by $NO_x$ over sources (e.g., the right part of Figure S1). However, evidence (Laughner and Cohen, 2019; Jin et al., 2020; Zhu et al., 2022) has emerged that many cities across the United States (US) recently experienced the transition into the $NO_x$-limited regime following emission regulations, where stronger $NO_x$ emissions actually promote $HO_x$

production and decrease $\tau$ (e.g., the left part of Figure S1). It is unclear how this change will update our understanding of the resolution dependency of simulated $NO_x$. Furthermore, nighttime effects were rarely studied possibly due to the prolonged $\tau$ and reduced $NO_x$ localization, although significant spatial heterogeneity of nocturnal $NO_x$ and ozone in urban environments were still noted (Zhang et al., 2004; Pan et al., 2017; Zakoura and Pandis, 2018), and long-term effects of changing $NO_x$ on nighttime $\tau$ were also evident (Shah et al., 2020). Finally, retrieval of tropospheric $NO_2$ column density ($\Omega$) from a growing

constellation of satellite instruments (e.g., OMI, TROPOMI, TEMPO, Sentinel-4, GEMS) offers observational information to constrain $NO_x$ emissions across vast continental and global regions (Veefkind et al., 2012; Streets et al., 2013; Zoogman et al., 2017; Levelt et al., 2018; Timmermans et al., 2019; Kim et al., 2020); Existing studies have not separately discussed the resolution effects on $\Omega$ and on surface $NO_x$, which can potentially differ due to the vertically nonuniform spatial heterogeneity and species abundances.

This study uses a CTM across a wide range of resolutions to significantly enrich current understanding of resolution dependency of $NO_x$ simulation. We find evidence over the eastern US that changes in simulated $NO_x$ abundances at different resolutions are by no means uniform, but depend on factors such as chemical regimes, dominant processes, and vertical layering. This information urges the necessity to apply adequate resolution to simulate and interpret $NO_x$-relevant atmospheric chemistry and air quality issues.



## 2 Materials and methods


We use the GEOS-Chem model in its high performance implementation (GCHP, http://www.geos-chem.org, version 13.2.1, DOI: 10.5281/zenodo.5500718) to simulate $NO_x$ and its relevant components over the eastern US.

GCHP is a grid-independent implementation of GEOS-Chem operating in a distributed-memory framework for massive parallelization (Long et al., 2015; Eastham et al., 2018). Chemical transport is simulated using a finite volume advection code

(FV3) on a cubed-sphere grid (Putman and Lin, 2007). GCHP uses identical chemistry and physics modules as the standard GEOS-Chem code (GEOS-Chem Classic). A stretched-grid capability offers finer resolution over a user-specified domain of interest (Bindle et al., 2021). The model version used here (v13.2.1) features significant advances for performance and ease of use (Martin et al., 2022). The model is driven by the Goddard Earth Observation System Forward Processing (GEOS-FP) assimilated meteorological data with the native resolution of $0.25° \times 0.3125°$, from the NASA Global Modeling and Assimi-

lation Office (GMAO). The GEOS-FP data is currently the finest resolution meteorology available for GCHP simulations for the simulation year, and was regridded to each simulation resolution, including the resolution (of 13 km) that is finer than $0.25° \times 0.3125°$. Although non-ideal, this capability as demonstrated by Bindle et al. (2021) will not significantly alter our interpretations focusing on discussing redistribution of $NO_x$ emissions and chemical feedbacks, rather than effects from meteorology. Yan et al. (2016) showed that sub-coarse-grid emission-chemical variability dominantly contributed to the differences

of simulated tropospheric chemistry between resolutions, overwhelming the effects from resolution of non-chemical factors such as meteorological data. Consistent with GEOS-FP, the atmosphere is vertically distributed into 72 layers (from surface to 0.01 hpa) following the hybrid sigma-pressure grid definition during the simulation. Boundary layer mixing is simulated with a non-local scheme (Lin and McElroy, 2010). The lowest layer is roughly 120 m thick, with mixing ratios of $NO_x$, $HO_x$ and ozone that we refer to as the "surface concentrations".

We use the standard full-chemistry scheme of the GEOS-Chem model which is widely used to study air quality (Koplitz et al., 2016; Li et al., 2017; Shah et al., 2020; Gu et al., 2021). The scheme includes detailed gas-phase mechanisms of $HO_x$-$NO_x$-VOC-ozone chemistry (Bey et al., 2001; Mao et al., 2013; Sherwen et al., 2016) including heterogeneous uptake of reactive gases (McDuffie et al., 2018; Holmes et al., 2019) by the simultaneously simulated aerosols. Anthropogenic $NO_x$ emissions are from EDGAR v5.0 at 0.1° resolution (Crippa et al., 2021), and speciated anthropogenic non-methane VOC emissions are

from CEDS v2 (Hoesly et al., 2018). Open burning emissions are from GFED v4.1 (van der Werf et al., 2017). Although the latter two inventories have non-ideal (0.5° and 0.25°) resolutions due to availability, they are acceptable for our purpose of identifying the resolution dependence of $NO_x$. One favorable capability of the simulation is the pre-calculated offline dust, lightning $NO_x$, biogenic VOC (BVOC), soil $NO_x$ and sea salt aerosol emissions (Murray et al., 2012; Weng et al., 2020; Meng et al., 2021), which avoids possible regional emission biases due to online calculations using meteorological fields at different

resolutions, and the consequent interference on the interpretation of the results. All the emissions are handled by the HEMCO 3.0 module (Keller et al., 2014; Lin et al., 2021).

We conduct GCHP simulations for January (winter) and July (summer) of 2015, at six resolutions spanning the range of conventional global model capabilities (Table 1). The highest resolution (13 km) is close to the currently finest information



from emission inventories (0.1°) in global CTMs without downscaling, and the lowest resolution (181 km) remains widely used

in global and regional air quality studies (i.e. similar to 2°×2.5° resolution). For the three relatively coarser resolutions, we conduct global cubed-sphere simulations, while for resolutions < 50 km, we exploited the recently developed grid-stretching capability (Bindle et al., 2021) to greatly reduce the computational resource requirements. The stretched-grid configuration smoothly decreases the grid cell size towards the refined region of interest. We follow recommendations by Bindle et al. (2021) to choose moderate stretch factors (Table 1), ensuring that the lowest resolutions (i.e. on the antipode of the refined region)

remain finer than 300 km whilst achieving the substantially refined resolutions over the eastern US (Table 1, right).

     Initial conditions on December 1, 2014 and June 1, 2015 are obtained from a 1-year spin-up run at C360 global cubed-sphere simulation using identical model version and inputs. These initial conditions resemble realistic global high-resolution (∼25 km) distribution of $NO_x$-relevant species and their oxidants driven by the consistent emissions and chemistry used in this study, and were then regridded to drive the 2-month simulations at each resolution. The second month (January and July 2015)

from each simulation is used for the analysis to allow for further spin-up at each resolution. We archive the hourly average model outputs to enable the detailed discussion in the result section.

     Figure 1a shows the study domain of interest (70–98°W and 26–48°N) and the emission ratio of isoprene to $NO_x$. The northern part of the domain comprises some of the most populous cities of the continent (e.g., New York, Toronto and Chicago), with strong and localized $NO_x$ emissions as observed from space (Russell et al., 2012; Lu et al., 2015; Laughner and Cohen,

2019), ideal for investigating $NO_x$ resolution dependence. The southern regime is characteristic of the strongest biogenic VOC (e.g., isoprene) emission rates across the US (Romer et al., 2016; Yu et al., 2016; Jin et al., 2020) that could affect the $NO_x$ lifetime and its response to resolution.

     To assess the implications for interpreting satellite observations of $NO_2$, we also apply monthly mean scattering weights ($w$) (Palmer et al., 2001) from the TROPOspheric Monitoring Instrument (TROPOMI) at its overpass time (UTC 19-21, namely

1-3 pm at central standard time (CST)) to calculate $NO_2$ line-of-sight (slant) column density ($\Omega_s$). Comparing $\Omega_s$ with $\Omega$ enables investigation of effects of satellite sensitivity on the resolution dependency of $NO_2$ columnar abundances. TROPOMI is currently the only instrument with sufficient fine resolution to provide $w$ information for all the investigated resolutions over the study domain. We assign every clear sky (i.e. geometric cloud fraction < 0.2) TROPOMI $w$ in January or July of 2019 to the collocated grid cells of each resolution, to derive the monthly mean $w$ distribution, which is then applied to the afternoon

mean $NO_2$ profiles in the same month of 2015 for each grid cell to calculate $\Omega_s$.

     In this study, all the regridding procedures between different resolutions are conducted using the conservative (surface area-weighted) algorithm (https://earthsystemmodeling.org/regrid/) in the Earth System Modeling Framework.

## 3   Results

### 3.1   Daytime resolution effects at surface in summer

Figure 1c shows the daytime (UTC 15-24, corresponding to CST 9-18) resolution effects of simulated surface $NO_x$ concentrations over the eastern US in July 2015. At the finest resolution of 13 km (upper left), $NO_x$ exhibits notable local enhancements





over cities and major industrial corridors due to its short lifetime ($\tau$, several hours). Overall, the stronger emissions and agglomerate sources in the Great Lakes region (GL, green box) lead to higher $NO_x$ levels than in the Southern States (SS, magenta box) where $NO_x$ sources are relatively weaker and sparser. As the grid cell size increases to 22 km resolution, the $NO_x$ level

shows overall decreases over emission centers and increases over nearby grids (by up to 1 ppb) relative to 13 km, an expected consequence due to dilution of emissions. However, systematic biases in predicted $NO_x$ relative to 13 km resolution start to emerge especially further downwind and over the SS, reflecting the effects from the resolution-dependent $\tau$. The biases relative to 13 km resolution become increasingly pronounced and regionally coherent as grid cells further enlarge. At the three coarse resolutions (> 50 km), a clear dipole of negative biases over the GL and positive biases over the SS becomes observable.

The opposite resolution effects of simulated $NO_x$ over the GL and SS are summarized as regional mean biases in each panel of Figure 1c (e.g., -17.5% at 181 km over the GL), which are attributable to their corresponding chemical regimes. The GL in summer is characterized by stronger $NO_x$ emissions and lower VOC emissions, while the opposite scenario prevails over the SS with strong BVOC sources, characterized partially by the distribution of isoprene/$NO_x$ emission ratio in Figure 1a. Consequently, the $NO_x$ sources in the GL tend to locate in the $NO_x$-saturated regime (Figure S1, right) where concentrated

$NO_x$ levels at higher resolutions consume more OH and increase $\tau$; Meanwhile over the SS, the relatively lower $NO_x$ together with the enhanced VOC can reversely promote $HO_x$ production at higher $NO_x$ levels ($NO_x$-limited regime, Figure S1, left), thus higher resolution will introduce higher OH and lower $\tau$. The lowered $\tau$ can be a result of directly scavenging $NO_2$ by the enhanced OH, and of indirectly sinking nitric oxide (NO) by the OH-promoted organic peroxy radicals ($RO_2$), an important $NO_x$ sink pathway under low-$NO_x$ and high-VOC environments (i.e. the SS) (Browne and Cohen, 2012; Perring et al., 2013;

Romer et al., 2016; Romer Present et al., 2020). As $NO_x$ emissions continue to decrease, multiple lines of evidence suggest that $NO_x$ sources widely across the United States have recently entered or are approaching the $NO_x$-limited regime (Laughner and Cohen, 2019; Jin et al., 2020; Koplitz et al., 2021; Jung et al., 2022; Zhu et al., 2022), especially over the SS since the 2010s (Jin et al., 2020; Koplitz et al., 2021; Zhu et al., 2022).

Figure 1b shows examples of daytime $HO_x$-$NO_x$ relationships for 12 cities. We use $HO_x$ to identify the regime represen-

tation since OH is unstable with a very short lifetime ($\sim$1s). $HO_x$ and $NO_x$ are anti-correlated over the GL (green labeled) while positively correlated over the SS (magenta labeled), consistently indicative of their different chemical regimes. The mean $NO_x$ concentrations (circles) in Figure 1b thus roughly decrease over the GL and increase over the SS following the opposite changes in $\tau$, as grid cell size increases (i.e. from magenta to brown circles). The mean $HO_x$ biases within the $2°\times2°$ windows above the cities are uniformly positive due to the opposite $HO_x$-$NO_x$ associations over the GL and SS.

Unlike over the GL where the regional biases of surface $NO_x$ gradually increase in magnitude following increasing grid cell sizes (Figure 1c, green numbers), the five resolutions > 13 km indicate small changes in the regional $NO_x$ concentration across the SS (i.e. biases of 6.4-8.7%); namely, the systematic increase occurs primarily between 13 and 22 km resolutions (Figure 1c, magenta numbers). This observation can be related with the variation of spatial extent of chemical regimes and their effects on the $NO_x$ biases during the course of the day. Figure S2 shows the resolution-dependence of simulated surface

$NO_x$ separately for morning (left) and afternoon (right). Relative to the overall effects during daytime (Figure 1c), the $NO_x$-saturated regime (with negative $NO_x$ biases at coarser resolution) has broader extent (e.g., intruding further into Tennessee) in





the morning hours, while the $NO_x$-limited regime (positive $NO_x$ biases at coarser resolution) can significantly affect southern Ohio during the afternoon. The magnitudes of $NO_x$ bias (e.g., 181 km vs. 13 km) are also enhanced over the GL in the morning (-27.1%) and over the SS in the afternoon (13.7%). These differences are consistent with the relative diurnal evolution of $NO_x$

(decreases since sunrise) and $HO_x$ (accumulates and peaks after noon) abundances and the consequence on the dominant $HO_x$ loss pathway (e.g., Ren et al., 2003; Ma et al., 2022). The inclusion of morning hours could therefore partially explain the relatively weaker resolution dependence in daytime $NO_x$ over the SS.

     Another potential cause of weaker sensitivity of simulated $NO_x$ to resolution is the impacts from BVOC in addition to $NO_x$ heterogeneity. Figure S3 shows that changing VOC reactivity mainly affects OH concentration and $\tau$ over the $NO_x$-

limited regime while has little effects on the $NO_x$-saturated regime, consistent with previous studies (Edwards et al., 2014; Laughner and Cohen, 2019; Zhu et al., 2022). Apart from increasing OH that decreases $\tau$ (Figure S3), decreasing VOC can also oppositely decrease the strength of the $NO$-$RO_2$ loss pathway and increase $\tau$ (Browne and Cohen, 2012). Nonetheless, both processes indicate increasing sensitivity of $\tau$ to VOC at low-$NO_x$ environments (Romer et al., 2016; Laughner and Cohen, 2019; Romer Present et al., 2020). The SS feature strong BVOC emissions as well as strong spatial segregation of $NO_x$ and

VOC sources (Yu et al., 2016; Travis et al., 2016), as reflected by lower isoprene/$NO_x$ emission ratios in urban centers relative to its neighborhood in Figure 1a. Such segregation is reduced as the resolution lowers (e.g., Figure S4). The concurrent and usually opposite changes of $NO_x$ (increases) and BVOC (decreases) emissions around $NO_x$ sources at higher resolutions can jointly lead to the overall small changes among resolutions > 13 km. In summary, our simulations reveal that the predictability of actual resolution dependency of $NO_x$ in the $NO_x$-limited regime is reduced due to the joint sensitivities to VOC.

## 3.2   Nighttime resolution effects at surface in winter

Figure 2a shows surface $NO_x$ concentration and its resolution dependence at nighttime (UTC 4-11 or CST 22-5) in January, 2015. With prolonged $\tau$ (~20 hours) as photochemistry ceases and OH becomes negligible, the wintertime and nighttime $NO_x$ exhibits reduced spatial heterogeneity (e.g., Figure S5) relative to summertime and daytime in Figure 1c, and the resolution effects are thus also less pronounced (i.e. ≤5%). However, one characteristic phenomenon at nighttime that depends on resolu-

tion is the titration between NO and ozone ($O_3$), which is the main nighttime sink to each other in urban environments (Brown et al., 2006; Wang et al., 2006; Brown and Stutz, 2012; Kenagy et al., 2018; Shah et al., 2020). Figures 2b and 2c indicate that the titration between NO and $O_3$ at the surface is enhanced by enlarged grid cells, as both concentrations were near uniformly reduced (by ~50% and ~10%, respectively) across the domain. At coarser resolutions, the faster titration produces more $NO_2$, complemented by less efficient scavenging of $NO_2$ by the more titrated $O_3$ (Figure 2d). The resolution effect on surface $NO_x$

(Figure 2a) is thus jointly contributed by the opposite changes in NO (Figure 2b) and $NO_2$ (Figure 2d), the latter being more determinant due to its stronger contribution to total $NO_x$.

     The faster titration efficiency at coarser resolution is driven by the spatial anti-correlation of NO and $O_3$, as demonstrated by Figure S6. Typical nighttime high-NO regions are coincident with low-$O_3$ locations at 13 km resolution (first row), a result of daytime $O_3$ formation suppression and nighttime $O_3$ titration (Jacob et al., 1995; Zhang et al., 2004; Jin et al., 2017; Yan et al.,

2018; Sicard et al., 2020; Li et al., 2022) over strong $NO_x$ sources. This anti-correlation at fine resolution leads to inefficient





NO-O$_3$ reaction, which is to first-order proportional to their products, shown in the third column in Figure S6. By simply diluting their concentrations to larger grid cells (2rd-6th rows), the products of NO-O$_3$ from less anti-correlated concentrations are enhanced systematically. Consequently, there would also be faster production of N$_2$O$_5$ and nitrates, which were proposed by Zakoura and Pandis (2018) to explain the systematic overprediction of nitrate aerosols by CTMs at coarse resolution.

As the GL region has greater NO concentrations than the SS (Figure 2b), surface O$_3$ is more effectively titrated (Figure 2c), leading to increased NO$_2$ and NO$_x$ concentrations (Figure 2d). Meanwhile over the SS, the NO$_2$ response is less pronounced due to the lower NO levels, and NO$_2$ exhibits reductions over certain locations at lower resolutions (Figure 2d), indicating that the excess O$_3$ can consume more NO$_2$ after titrating NO over these grids.

### 3.3 Seasonal and diel variation of relevant processes

Figure 3 summarizes the resolution effects of regional mean surface NO$_x$, HO$_x$ and ozone, at different hours of the day. Over the GL, the strongest percentage biases of regional mean NO$_x$ (mainly NO$_2$, Figure S7) at resolutions > 13 km occur at nighttime in January (up to 5%), and at daytime in July (up to -30%), revealing a pronounced seasonality of dominant mechanisms driving the resolution effects. This seasonality is driven by the stronger intensity and duration of daytime oxidant production in July (i.e. magenta lines for HO$_x$ and O$_3$ in Figure 3); meanwhile the greater nighttime O$_3$ titration at coarser 205 resolution partially counteracts the daytime effects in July (i.e. reduces the percentage biases) and dominates in January.

The SS region has relatively stronger daytime resolution sensitivity that compensates the opposite nighttime effects on NO$_2$ in January (Figure S7), resulting in overall small changes in NO$_x$. In July, the resolution effects are again dominated by the daytime processes but in the NO$_x$-limited regime, which reverse the titration-driven opposite nighttime effects (more notable for HO$_x$ and O$_3$). Unlike over the GL or in January, NO becomes remarkably determinant of the diel variation of NO$_x$ biases 210 (Figure S7), reflecting the importance of the NO+RO$_2$ loss pathway on $\tau$ over the summer of the SS (Browne and Cohen, 2012; Perring et al., 2013; Romer et al., 2016; Romer Present et al., 2020). The small resolution dependence of the regional NO/NO$_x$ biases at resolutions > 22 km again reflects the joint and compensating effects on NO$_x$ lifetime from NO$_x$ localization and VOC segregation (Section 3.1).

Overall, the daytime resolution effects driven by the involvement of NO$_x$ in HO$_x$ and O$_3$ production (Section 3.1) compete 215 with the nighttime effects driven by NO$_x$-O$_3$ titration (Section 3.2). The changing dominance of each mechanism during summer vs. winter, as well as during daytime and nighttime, leads to the characteristic seasonal and diel variation in Figure 3.

### 3.4 Vertically variable resolution effects

Figure 4 shows the resolution-dependent changes in regional mean daytime NO$_x$ vertical profile in the lower troposphere (below 4 km). Uniform decreases in the simulated daytime NO$_x$ following larger grid cells are apparent at > 1km altitude in 220 July, despite opposite changes over the two regions near surface (Figure 1). These vertically dependent responses are caused by the different vertical profiles of NO$_x$ and HO$_x$ (i.e. magenta lines). As NO$_x$ mixing ratio decreases exponentially aloft while HO$_x$ increases (in the GL) or remains relatively uniform (in the SS), HO$_x$ becomes more abundant relative to NO$_x$ at higher altitudes, meaning that $\tau$ is less sensitive to NO$_x$ local concentrations even above strong NO$_x$ sources. The enhanced oxidants





(ozone and $HO_x$) due to surface $NO_x$ emission heterogeneity (Section 3.1) then vertically mixed to systematically enhance

$HO_x$ profile (Figure 4, right) and reduce $\tau$ and $NO_x$ in these aloft layers. Therefore, both regions exhibit negative $NO_x$ biases due to coarse resolution above 1 km, regardless of chemical regime.

Figure 5 shows the changes in nighttime vertical profile of $NO_x$ and $O_3$ in January. Again, there are opposite vertical distributions of $NO_x$ and its nighttime sink (ozone). Over the GL, although surface $NO_x$ lifetime can be possibly prolonged at coarse resolution due to the faster titration of $O_3$ by NO (Section 3.2), NO quickly becomes insufficient to titrate the

increasing ozone at higher altitudes. Therefore, both $NO_x$ species ultimately become affected by the resolution-dependent titration efficiency above 1 km (similar to the surface responses over the SS), leading to the negative biases in simulated $NO_x$, regardless of surface $NO_x$ emission strength.

In summary, Figures 4 and 5 reveal that the resolution effects of $\tau$ at surface can differ from those at elevated altitudes, even over source regions. Such altitude-dependent responses will further affect interpretation of satellite-retrieved $NO_2$ columnar

properties, using model simulations at these resolutions.

### 3.5    Implications for satellite remote sensing applications

Satellite retrievals of $NO_2$ vertical column density ($\Omega$) have been widely used to quantify and characterize spatiotemporal variation of $NO_x$ abundances and sources. Here we evaluate the implications of the $NO_x$ resolution dependency on two major applications—estimating surface $NO_2$ concentration and deriving $NO_x$ emissions.

Figure 6 shows the simulated fraction of $NO_2$ abundance within the surface layer of GEOS-Chem relative to the whole troposphere, during Low Earth orbit (LEO) afternoon satellite overpass time. This surface fraction ($F_s$) is lower in summer than in winter, driven by stronger convection, lightning $NO_x$, and elevated boundary layer. $F_s$ is also lower over the SS than the GL in July, due to relatively less $NO_x$ emissions at surface and stronger lightning $NO_x$ emissions in the upper troposphere (Murray et al., 2012; Silvern et al., 2019; Zhu et al., 2019). The changes of $F_s$ following varying resolutions in general qualitatively

resemble these of surface $NO_x$ (e.g., comparing Figure 6 with Figures 1c and Figure 2a). Overall, $F_s$ has stronger biases in July than in January, lowered by 8% over the GL and enhanced by 7% over the SS at 181 km, relative to 13 km resolution. As $F_s$ is a key parameter in estimating surface $NO_2$ concentration from satellite retrieved $\Omega$ (Lamsal et al., 2008; Cooper et al., 2020), directly applying the simulated $NO_2$ vertical profiles will propagate such resolution-dependent biases that also vary regionally and seasonally, as indicated in Figure 6.

Figure 7 shows the simulated tropospheric $NO_2$ vertical column density ($\Omega$) and its resolution dependence during LEO afternoon overpass time, and Figure S8 shows the corresponding slant column density ($\Omega_s$) after applying TROPOMI scattering weights. Relative to the surface biases (Figure 1c), the $\Omega$ and $\Omega_s$ differences show stronger regional uniformity, revealing overall increasingly negative biases with increasing grid size over both seasons and regions. For July and over the SS, the daytime columnar biases are reversed to be negative compared to the positive surface biases (e.g., Figure 7 vs. Figure 1c), driven by the

reversed responses at higher altitudes (Figure 4). This remarkable reversal of $\Omega$ biases (negative) vs. surface biases (positive) reinforces the need to include vertical profile information in correctly simulating $NO_x$ columnar properties. $\Omega$ and $\Omega_s$ exhibit quantitatively consistent resolution dependence and its spatial distribution (Figure 7 vs. Figure S8), indicating that the changes





of vertical profile do not significantly alter the resolution dependency of satellite observed columnar abundances, compared to the direct effects from changing $\tau$. Assuming $NO_x$ emissions are locally related with $NO_2$ columns at coarse resolution will result in similar magnitudes of overprediction (up to 10% and 7% over the GL and SS, respectively in July) of derived $NO_x$ emissions to compensate the underestimated $\Omega$ to emission relationship, under an inverse modeling framework configured at each resolution, as shown in Figures 7 and S8.

## 4 Discussions and conclusions

The strong, and more importantly regionally variable, $NO_x$ resolution dependences that we find in the lower troposphere over the contemporary eastern US warrants care in interpreting coarse resolution $NO_x$ simulations. We find that resolution-dependent $NO_x$ biases are particularly large at the surface in summer, with variable effects seasonally, regionally, diurnally, and vertically, which also affect remote sensing observations interpreted with low resolution simulations. Existing literature about $NO_x$ resolution dependencies in box models (Valin et al., 2011), in power plant (Sillman et al., 1990) and ship plumes (Charlton-Perez et al., 2009), and in CTMs (Wild and Prather, 2006; Yamaji et al., 2014; Yan et al., 2016) primarily discusses the $NO_x$-saturated regimes; We find limited prior literature about the positive biases of $NO_x$ over weak sources (i.e. in the $NO_x$-limited regime over the SS) in CTM simulations. The lack of similar prior reports reflects the chemical regime transition occurring in the recent ~10 years, while previously, typical point sources were predominantly in the $NO_x$-saturated regime. Attention to the $NO_x$-limited regime and its corresponding resolution effects is timely given declining $NO_x$ emissions across the US with $NO_x$ emission regulations. At the same time, the joint sensitivity to $NO_x$ heterogeneity and concurrent VOC level (Section 3.1) in the $NO_x$-limited regime will continue complicating its predictability, since $NO_x$ and VOC can have various spatial co-variabilities (e.g., positively correlated where transportation-relevant VOC and $NO_x$ both dominates) and regime-dependent effects on $\tau$. Therefore, accurately capturing such regime difference and transition from CTM requires not only accurate emission inventories of $NO_x$ and VOC, but also simulations at representative spatial scales that correctly distribute these emissions.

We found systematic resolution effects of nighttime $NO$-$O_3$ titration efficiency that can drive the $NO_x$ biases over winter (Figure 2 and Section 3.2), as the anti-correlation between $NO$ and $O_3$ implies faster reaction rates at coarser resolutions. In air quality modeling, many key reactions involve spatially correlated (e.g., co-emitted $SO_2$ and $NO_2$ to cause severe urban haze (Wang et al., 2020)) or segregated species (e.g., agricultural $NH_3$ and $NO_x$-formed $HNO_3$ for nitrate aerosol partitioning (Gu et al., 2021)). Like in this study, the segregated species will consume precursors and produce products more efficiently at coarser resolutions, while collocated sources will experience opposite effects. Interpreting the evolution of relevant species and air pollution processes using CTM is therefore also preferable at the spatial scales that are representative of these sources, or should take this effect into account if performed on coarser scales.

Our detailed simulation of resolution effects at different altitudes (Figures 4 and 5) significantly enriched the understanding of resolution dependency of satellite columnar observations, in contrast to previous studies that neglected vertical layering. The example of opposite resolution effect of $\Omega$ and surface $NO_x$ (Figure 7 vs. Figure 1c) over the SS and July highlights the





necessity of realistic vertical profiling of $NO_x$ and its chemical sinks. For two conventional applications of satellite retrieved $\Omega$, namely estimation of surface $NO_2$ concentration and constraining $NO_x$ emissions, we found that regionally and seasonally varying biases at the level of $\sim$10% due to adopting coarse model simulations ($\sim$200 km) are inevitable.

Overall, we conducted a comprehensive novel evaluation of $NO_x$ resolution dependence using a CTM across a wide range
of resolutions (13-181 km) and scenarios (including nighttime, winter and higher altitudes). We found the strongest resolution effects in the summer and daytime (e.g., -18% for surface $NO_x$ and -10% for columnar $NO_2$ over the Great Lakes), where and when the $NO_x$ spatial heterogeneity is the strongest and its lifetime is the shortest (e.g., Figure S5). These systematic resolution dependences should be considered when constraining model parameters (e.g., emissions, reaction yields, removal rates, etc.) using ground- or satellite-based observations. In other words, relevant interpretations and conclusions by coarse
model simulations in previous studies are worth revisiting.

Although this study exploited state-of-science capabilities, biases with respect to resolutions finer than 13 km resolution likely exist considering the severely localized $NO_x$ especially in summer (Valin et al., 2011; Larkin et al., 2017; Beirle et al., 2019). Following $NO_x$ regulations in the US, the magnitudes of resolution effects are expected to continue decreasing as the enhancements over sources reduced relative to the background $NO_x$ level (Russell et al., 2012; Jin et al., 2020; Qu et al.,
2021), and the requirements on resolution may diminish (e.g., partially reflected by the smaller effects over the SS relative to over the GL). Nonetheless, over developing areas where current $NO_x$ emissions are stronger or are projected to increase, the resolution effects will be exacerbated, and applying finer-resolution simulations to accurately capture $NO_x$ lifetime and budgets will be increasingly critical for air quality modeling applications. Optimization of appropriate resolution that can capture the relevant processes accurately for specific applications given computational resource constraints is also of great interest. GCHP
offers the unique global high resolution simulation capability, and also the opportunities to expand this analysis into a more comprehensive understanding of global resolution-dependence of $NO_x$ and its nonlinear chemistry.

*Code and data availability.* GEOS-Chem 13.2.1, including GCHP, is available for download at https://doi.org/10.5281/zenodo.5500718 (The International GEOS-Chem User Community, 2021). TROPOMI $NO_2$ data is available from https://tropomi.gesdisc.eosdis.nasa.gov/ data/S5P_TROPOMI_Level2. The hourly model output for two months and six resolutions are available upon request to the corresponding
author (chili@wustl.edu; lynchlee90@gmail.com)

*Author contributions.* The manuscript was written through contributions of all authors. The conceptualization was initialized by CL, RVM and RCC. The methodology is developed by CL, LB and DZ. DC processed the satellite scattering weights. HW and JL conducted the offline emission calculation. CL performed the model simulations, visualization and analysis of the results. CL wrote the original draft. All authors have reviewed, edited, and given approval to the final version of the manuscript.



*Competing interests.* Ronald Cohen is a member of the editorial board of Atmospheric Chemistry and Physics. The contact author has declared that none of the other authors has any competing interests.

*Acknowledgements.* This work is supported by the NASA AIST (80NSSC20K0281) and ACCDAM (80NSSC21K1343) programs. We thank the GEOS-Chem support team for maintaining the feasibility of model simulations in this work.





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



**Figure 1. Daytime (UTC 15-24 or CST 9-18) resolution effects of surface $NO_x$ concentration driven by $NO_x$-$HO_x$ feedbacks** (e.g., Figure S1) over July, 2015. (a) Emission ratio of isoprene vs. $NO_x$; (b) Simulated $HO_x$-$NO_x$ relationship at each resolution (color-coded). The contour lines (for resolutions < 60 km) include 90% of the 2-d scatter (points) of data within 2°×2° window centered over each city (based on kernel density estimation), and the circles place the means for all resolutions; (c) Surface $NO_x$ concentration at 13 km (top left) and the differences (minus 13 km) at the other resolutions (other panels). The regional mean $NO_x$ (for 13 km, in ppb) and percentage biases (for the other resolutions) are indicated at the bottom right of each panel. Green and magenta (rectangles, city names and numbers) label the Great Lakes region and Southern States, respectively.





**Figure 2.** **Nighttime (UTC 4-11 or CST 22-5) resolution effects of surface** $NO_x$ **concentration driven by** NO **titration of** $O_3$ (e.g., Figure S6) during January, 2015. Each panel is similar to Figure 1c but for nighttime surface concentrations of (a) $NO_x$, (b) NO, (c) $O_3$, and (d) $NO_2$.





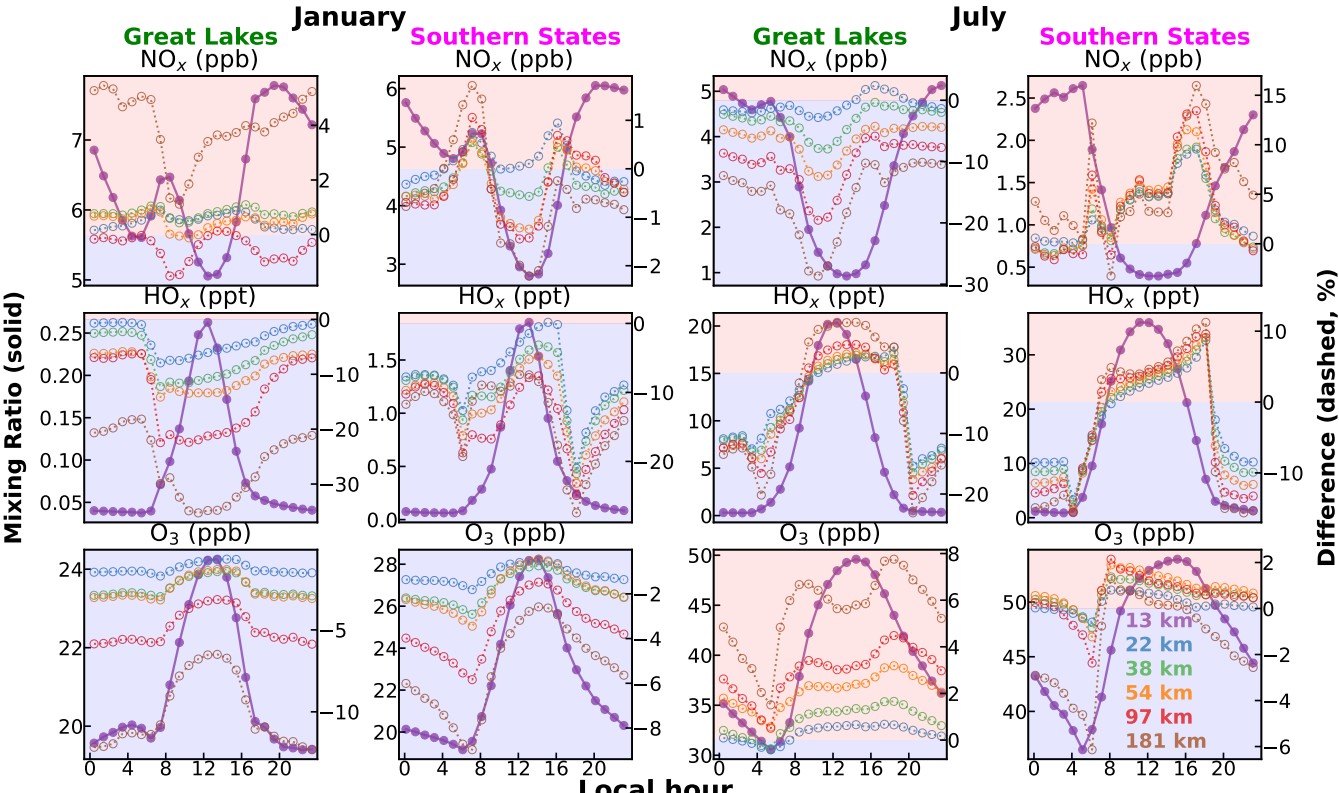

**Figure 3. Resolution effects of surface** $NO_x$ **dominated by nighttime biases in January and by daytime biases in July**, 2015. The absolute monthly mixing ratios (left Y-axis) of regional mean $NO_x$, $HO_x$ and $O_3$ are shown as filled symbols and solid lines at 13 km resolution, and the differences vs. 13 km (right Y-axis) are shown as empty symbols and dashed lines for the other resolutions (see color-legends). The positive (light red) and negative (light blue) difference regimes are color-divided.





**Figure 4. Change of daytime $NO_x$ vertical distribution in the lower troposphere at different resolutions** in July. The absolute monthly and regional mean concentrations (lower-X axis) of $NO_x$ and $HO_x$ are shown as filled symbols and solid lines at 13 km resolution, and the differences vs. 13 km (upper-X axis) is shown as empty symbols and dashed lines for the other resolutions (see color-legends). The positive (light red) and negative (light blue) difference regimes are color-divided.



**Figure 5. Change of nighttime NO$_x$ vertical distribution in the lower troposphere at different resolutions** in January. Similar to Figure 4 but for nighttime NO$_x$ and O$_3$ in January.





Fraction of tropospheric NO$_2$ column within the surface layer at 13 km and biases at the other resolutions

(a) January

(b) July

**Figure 6. Coarse resolution simulations yield variable biases in satellite-based estimation of surface NO$_2$ concentration.** Panels (a) and (b) are both similar to Figure 1c (for January and July, 2015, respectively) but for the fraction (%) of NO$_2$ tropospheric column within the surface layer during afternoon satellite overpass time (UTC 19-21). The numbers for the other resolutions vs. 13 km are absolute differences in percentage number.





**Figure 7. Coarse resolution simulations yield positive biases in spaceborne inverse modeling of** $NO_x$ **emissions**. Panels (a) and (b) are both similar to Figure 1c (for January and July, 2015, respectively) but for the mean GEOS-Chem tropospheric $NO_2$ column density (in molecules/cm$^2$) during afternoon satellite overpass time (UTC 19-21).



**Table 1.** Description of the simulations with six resolutions over the eastern US.

| Cubed-sphere grids[a] | Stretch factor[b] | Center of the refined domain | Average resolution in the eastern US (km) |
|---|---|---|---|
| C48 | | | 181 |
| C90 | N/A | N/A | 97 |
| C160 | | | 54 |
| C100 | 2.8 | | 38 |
| C136 | 3.5 | 37°N, 84°W | 22 |
| C180 | 4.3 | | 13 |

[a] A cubed-sphere grid contains a mosaic of six grids (faces). Each face is regularly spaced with N×N grid cells, and the notation of each resolution (CN) here identifies the size N.

[b] Stretch factor (S) defines the strength of the grid-stretching. The resolution is about S times higher than the regular cubed-sphere over the refined region (eastern US), while is about 1/S of the original resolution over the antipode.