# Peer review of "Variable effects of spatial resolution on modeling of nitrogen oxides"

_EGUsphere, 2022_

## Referee Comment (RC1)

Li et al. shows in their study, that the choice of the spatial model resolution can significantly influence the results of simulations using the GEOS-Chem model. They performed simulations with six different spatial resolutions for December 2014 to January 2015 and for June to July 2015 show the results from $NO_x$, $HO_x$, and $O_3$ for the eastern United States. The influence of the resolutions depends on whether high or low levels of $NO_x$ prevail in an area. For example in the Southern States where usually rather low $NO_x$ concentrations prevail, the simulations with smaller resolutions tends to an increase in NOx, whereby in the area of Great Lakes, where usually rather high $NO_x$ concentrations prevail, the simulations with smaller resolutions simulate smaller values of $NO_x$. The authors also show the corresponding influence on $HO_x$ and $O_3$ in these areas.

The $NO_x$, $HO_x$ and $O_3$ results are discussed in detail. The authors show that the spatial model resolutions has not only an influence on the surface, but also on higher altitudes. It also has an effect regarding the diurnal and seasonal cycle of the mentioned substances.

**General comments:**

In my opinion, the scientific relevance of this study is very high, because the authors show the choice of the spatial resolution influence the result of their model significantly. That means that different resolutions leads to different results. If the explanation of the reasons for this model behaviour are true, then this should be the case with every other model. This would have a high impact to every model community.

However, exactly the discussion regarding the reasons, which explain the different results with different resolutions isn´t clearly demonstrated in the paper. How I see this, one reason is only mentioned in the caption of Fig. S1 in the Supplement (!) and sometimes only between the lines. Here the reason is described, that higher resolution modelling tends to concentrate the most $NO_x$ emissions near sources. I think the discussion of this and maybe other reasons should really be more prominent in the main text and the reasons should also be mentioned in the abstract and in the discussion. Because the reasons are really important also for other modellers. If the know them, they can then also try this out with their own model.

I´m also thinking about, whether really only the point sources of the emissions are responsible or are there perhaps also other reasons which are influence the results (e.g. transport processes) or maybe different time steps in the used simulations with different spatial resolutions, which maybe affect the chemistry? It should be clear which possibilities exist and how confident the authors are that this possibilities really lead to the different model behaviour.

Another important point for me are the figures of the supplement, and the supplement itself. In the paper the discussion relating the figures generally jumps between the figures from the main section and the figures of the supplement. In my opinion, this is not convenient in this form. In my eyes, a supplement should only be a supplement. If a figure is important for the explanations in the main text, then it should also be in the main text. The explanation of the not so important figures of the supplement should be written in the supplement. The main text can mention that figures in the supplement exist, but not much more. For example, the figures S1 and S3 are in my eyes really essential to understand the content of the paper. Therefore, these two figures should be in any case in the main text. I would also set figures S2 and S8 to the main text and leave only figures S4 to S7 in the supplement. But of course the authors have to decide this.

What also comes in my mind, I have the feeling that the authors always assume that the highest resolution gives the best results, but is that a sure thing? Maybe the simulation with e.g. 38 km has the best results in comparison to observations. Are there comparisons with measurement data

possible? Can you from the comparisons with TROPOMI conclude which simulation is closest to reality?

Overall, I can absolutely recommend this paper for publication in ACP, in the case that there is a more detailed discussion with regard to the explanation what the reasons are, that the spatial model resolutions has such a high influence to the $NO_x$ (and $HO_x$ and $O_3$) mixing ratios.

**Specific comments:**

Throughout the paper:

- You always write of concentrations but I think you are referring most time to mixing ratios, please change this.
- In the case that you use ppt or ppb please use instead pptv or ppbv
- Change "Figure" to "Fig" and "Figures" to "Figs"

Main text:

Page 1, line 1: "The lifetime and concentration of nitrogen oxides ($NO_x$) are susceptible to non-linear production and loss, and consequently to the resolution of a chemical transport model (CTM)." I am not sure if you can really say here "consequently".

Page 1, line 2: "Here we use" → "In this study we use"

Page 1, line 3: "resolutions" → "spatial model resolutions" and "(13-181 km)" →"(six different horizontal grids from 13 to 181 km)"

Page 1, line 4 to 5: Please mention that the GL region is a $NO_x$-saturated area and the SS is a $NO_x$-limited area. Not every reader know this at this point of the paper.

Page 1, line 5: I would write here "differences to highest resolution of 13 km" instead of "biases" and "-18% to 9%" → "-18% (in GL region) to 9% (in SS region)"

Page 1, line 14 to 15: As I already said, it would be good also to mention the reason for this model behaviour already in the abstract.

Page 1, line 17 and 18: " $\equiv$ " → "="

Page 2, line 29: "outstanding" →"unexplained"

Page 2, line 33: "modeled" → "simulated"

Page 2, line 40: I would explain Figure S1 here a little bit more and move the figure from the Supplement to the main text.

Page 2, line 40 and line 50: "resolution" → "spatial resolution"

Page 2, line 50: Do you also use different time steps using the different spatial resolutions? If so, I think that could also have an impact on the chemistry and the NOx results.

Page 2, line 54: Is the "adequate resolution" always the highest resolution?

Page 4, line 104: "from space". From which satellite instruments?

Page 4, line 108 to 115: You use here the NO2 satellite observations from TROPOMI of the year 2019. Why do you choose the year 2015 for your simulations and not the year 2019?

Page 4, line 120: "concentrations" →"mixing ratios"

Page 4, line 121: "(upper left)" → "(Fig. 1c, upper left)"

Page 5, line 127: Where is "further downwind"?

Page 5, line 129: "observable" → "visible"

Page 6, line 163: Here you write "Another potential cause of the weaker", but you don't really explain the first potential cause before (at least in my opinion, except for the caption of Fig. S1).

Page 6, line 184 and line 187: "faster titration" → "faster O3 titration"

Page 7, line 209 and line 216: "diel" → "diurnal"

Page 8, line 250-251 and line 257 and line 262: You compare here one figures from the main part and one figure from the supplement. I would really transfer Fig. S8 to the main part.

Page 9, line 278: Means that you should use the highest possible model resolution?

Page 10, line 300ff: Here I would add a paragraph explaining the reasons for the different results of the simulations with different resolutions.

Figures:

Fig.1:

- Not in every panel of Fig 1c are the green and magenta boxes are drawn. Please change this
- Please use ppbv and pptv instead of ppb and ppt

Fig.2:

- Also here the green and magenta boxes are not drawn in every panel.

Fig. S1:

- "concentrations" → "mixing ratios"
- "ppt" → "pptv"; "ppb" → "ppbv"

Fig. S3:

- "ppt" →"pptv"; "ppb" →"ppbv"

---

## Author Comment (AC1)

We would like to thank both reviewers for their useful comments and suggestions that significantly helped improve the quality of this manuscript. We have addressed the comments (italicized) with point-by point responses (blue) provided below. All the Figures, Section and Line numbers refer to the revised manuscript (without tracked revisions).

*Reviewer 1*

*Li et al. shows in their study, that the choice of the spatial model resolution can significantly influence the results of simulations using the GEOS-Chem model. They performed simulations with six different spatial resolutions for December 2014 to January 2015 and for June to July 2015 show the results from NOx, HOx, and O3 for the eastern United States. The influence of the resolutions depends on whether high or low levels of NOx prevail in an area. For example in the Southern States where usually rather low NOx concentrations prevail, the simulations with smaller resolutions tends to an increase in NOx, whereby in the area of Great Lakes, where usually rather high NOx concentrations prevail, the simulations with smaller resolutions simulate smaller values of NOx. The authors also show the corresponding influence on HOx and O3 in these areas.*

*The NOx, HOx and O3 results are discussed in detail. The authors show that the spatial model resolutions has not only an influence on the surface, but also on higher altitudes. It also has an effect regarding the diurnal and seasonal cycle of the mentioned substances.*

*General comments:*

*In my opinion, the scientific relevance of this study is very high, because the authors show the choice of the spatial resolution influence the result of their model significantly. That means that different resolutions leads to different results. If the explanation of the reasons for this model behaviour are true, then this should be the case with every other model. This would have a high impact to every model community.*

Reply: We appreciate these positive comments on the manuscript. Using our results to inform the community of air quality modeling and its applications about the significance of model resolution is exactly the core motivation and implications delivered from this work.

*However, exactly the discussion regarding the reasons, which explain the different results with different resolutions isn´t clearly demonstrated in the paper. How I see this, one reason is only mentioned in the caption of Fig. S1 in the Supplement (!) and sometimes only between the lines. Here the reason is described, that higher resolution modelling tends to concentrate the most NOx emissions near sources. I think the discussion of this and maybe other reasons should really be more prominent in the main text and the reasons should also be mentioned in the abstract and in the discussion. Because the reasons are really important also for other modellers. If the know them, they can then also try this out with their own model.*

Reply: We added/revised clear discussions and summaries at noticeable places in the revised manuscript, about the main mechanisms driving the $NO_x$ resolution-dependence. These revisions better motivate and streamline the manuscript:

Abstract Line 1-3: "The lifetime and concentration of nitrogen oxides ($NO_x$) are susceptible to non-linear production and loss, and to the resolution of a chemical transport model (CTM), due to the strong spatial gradients of $NO_x$ and the dependence of its own chemical loss on such gradients."

Line 34-37 in the Introduction section: "Systematic differences in the simulated $\tau$ and $NO_x$ concentration at different resolutions were reported from simulated $NO_x$ plumes of power plants, cities and ship emissions (Sillman et al., 1990; Charlton-Perez et al., 2009; Valin et al., 2011), due to the stronger $NO_x$ localization at higher resolution."

Line 53-55 in the Introduction section: "Given that numerous existing studies depicted an evident yet potentially incomplete mechanistic understanding, this study uses a CTM across a wide range of spatial resolutions to significantly enrich current understanding of resolution dependency of $NO_x$ simulation."

Line 274-276 in the Discussion section: "At daytime with strong photochemistry, higher resolution modeling more realistically concentrates $NO_x$ emissions near sources, thus decreasing $\tau$ in the $NO_x$-limited regime and increasing $\tau$ in the $NO_x$-saturated regime (Fig. A1)."

Line 289-290 in the Discussion section: "We found systematic resolution effects of nighttime $NO$-$O_3$ titration efficiency that can drive the $NO_x$ biases over winter (Fig. 2 and Section 3.2), as the anti-correlation between $NO$ and $O_3$ implies faster reaction rates at coarser resolutions."

Line 297-299 in the Discussion section: "Our detailed simulation of resolution effects at different altitudes (Figs. 4 and 5) revealed vertically variable sensitivity of $NO_x$ to its chemical loss at different spatial resolutions. These findings significantly enriched the understanding of resolution dependency of satellite columnar observations, in contrast to previous studies that neglected vertical layering."

We also note key discussions in the Result section that clearly present the driving mechanisms in the manuscript:

Line 144-147: "... NOx sources in the GL tend to locate in the $NO_x$-saturated regime (Fig. A1a, right) where concentrated $NO_x$ levels at higher resolutions consume more OH and increase $\tau$; Meanwhile over the SS, the relatively lower $NO_x$ together with the enhanced VOC can reversely promote $HO_x$ production at higher $NO_x$ levels ($NO_x$-limited regime, Fig. A1a, left), thus higher resolution introduces higher OH and lower $\tau$."

Line 191-195: "...titration between NO and $O_3$ at the surface is enhanced by enlarged grid cells, as both concentrations were near uniformly reduced (by up to ~50% and ~10%, respectively) across the domain. At coarser resolutions, the faster $O_3$ titration produces more $NO_2$, complemented by less efficient scavenging of $NO_2$ by the more titrated $O_3$ (Fig. 2d). The resolution effect on surface NOx (Fig. 2a) is thus jointly contributed by the opposite changes in NO (Fig. 2b) and $NO_2$ (Fig. 2d), the latter being 195 more determinant due to its stronger contribution to total $NO_x$."

Line 199-202: "This anti-correlation at fine resolution leads to inefficient NO-$O_3$ reaction, which is to first-order proportional to their products, shown in the third column in Fig. A4. By simply diluting their concentrations to larger grid cells (2nd-6th rows), the products of NO-$O_3$ from less anti-correlated concentrations are enhanced systematically."

Line 221-223: "Overall, the daytime resolution effects driven by the involvement of $NO_x$ in $HO_x$ and $O_3$ production (Section 3.1) compete with the nighttime effects driven by $NO_x$-$O_3$ titration (Section 3.2). The changing dominance of each mechanism during summer vs. winter, as well as during daytime and nighttime, leads to the characteristic seasonal and diel variation in Fig. 3."

Line 227-232: "These vertically dependent responses are caused by the different vertical profiles of $NO_x$ and $HO_x$ (i.e. purple lines). As $NO_x$ mixing ratio decreases exponentially aloft while $HO_x$ increases (in the GL) or remains relatively uniform (in the SS), $HO_x$ becomes more abundant relative to $NO_x$ at higher altitudes, meaning that $\tau$ is less sensitive to $NO_x$ local mixing ratios even above strong $NO_x$ sources. The enhanced oxidants 230 (ozone and $HO_x$) due to surface $NO_x$ emission heterogeneity (Section 3.1) then vertically mix to systematically enhance the $HO_x$ profile (Fig. 4, right) and reduce $\tau$ and $NO_x$ in these aloft layers."

Line 234-239: "Again, there are opposite vertical distributions of $NO_x$ and its nighttime sink (ozone)...NO quickly becomes insufficient to titrate the increasing ozone at higher altitudes. Therefore, both $NO_x$ species ultimately become affected by the resolution-dependent titration efficiency above 1 km (similar to the surface responses over the SS), leading to the negative biases in simulated $NO_x$, regardless of surface $NO_x$ emission strength."

*I'm also thinking about, whether really only the point sources of the emissions are responsible or are there perhaps also other reasons which are influence the results (e.g. transport processes) or maybe different time steps in the used simulations with different spatial resolutions, which maybe affect the chemistry? It should be clear which possibilities exist and how confident the authors are that this possibilities really lead to the different model behaviour.*

Reply: Thanks for this constructive comment.

We have repeated our simulations by adopting consistent time step settings across all resolutions.

Line 100-102: "In this work, we apply chemical operator durations of 20 minutes and transport operator durations of 10 minutes (i.e., C20T10) (Philip et al., 2016) to all the simulations, largely in accordance with the GCHP default for cubed-sphere simulations while avoiding interferences to our interpretation from inconsistent operator duration settings."

The revised results and conclusions are overall consistent with the original manuscript (e.g., see the overall unchanged discussion text in the Result Section). One moderate deviation is the reduced temporal extent (CST 13-18 instead of 9-18) and magnitude (e.g., 7% at 181 km instead of 9%) of locations of $NO_x$-limited regime in the Southern States (e.g., see the revised Fig. 1 in the Tracked-change document). These more sensitive changes relative to other results are consistent with our discussion (Line 163-183) that the existence of $NO_x$-limited regime in the Southern States is sensitive to not only $NO_x$ spatial heterogeneity but also variability of other factors such as VOC and oxidation environment.

Other non-emission and non-chemistry factors such as transport might also exist, but they are fully coupled with the chemistry-driven feedbacks we interpreted in this paper, thus are hard to disentangle. We added in Line 307-310 of the Discussion Section a clarification of this issue:

"We attribute these systematic simulation biases mainly to the strong localization of $NO_x$ emission and chemistry at a spatial scale of ~10 km and less. Additional modulations from other factors across resolutions, such as sub-grid meteorology-relevant processes (e.g. transport) are also possible, but are fully coupled with the feedbacks revealed in this paper and are non-trivial to disentangle."

We also note in Line 71-75 existing discussion about other factors (including meteorology) besides emissions and chemistry:

"...this capability as demonstrated by Bindle et al. (2021) will not significantly alter our interpretations focusing on discussing redistribution of $NO_x$ emissions and chemical feedbacks, rather than effects from meteorology. Yan et al. (2016) showed that sub-coarse-grid emission-chemical variability dominantly contributed to the differences of simulated tropospheric chemistry between resolutions, overwhelming the effects from resolution of non-chemical factors such as meteorological data."

*Another important point for me are the figures of the supplement, and the supplement itself. In the paper the discussion relating the figures generally jumps between the figures from the main section and the figures of the supplement. In my opinion, this is not convenient in this form. In my eyes, a supplement should only be a supplement. If a figure is important for the explanations in the main text, then it should also be in the main text. The explanation of the not so important figures of the supplement should be written in the supplement. The main text can mention that figures in the supplement exist, but not much more. For example, the figures S1 and S3 are in my eyes really essential to understand the content of the paper. Therefore, these two figures should be in any case in the main text. I would also set figures S2 and S8 to the main text and leave only figures S4 to S7 in the supplement. But of course the authors have to decide this.*

Reply: We appreciate the careful evaluation of each figure from the reviewer, and we moved several figures (Figs. S1, S2, S3, S6, S8) originally in the supplement but are more elaborately discussed in the main text, into the Appendix A (Figs. A1, A3-A5). In this way, the different emphasis is conserved while the more important figures can be easily found in the same document without referring to the supplement.

*What also comes in my mind, I have the feeling that the authors always assume that the highest resolution gives the best results, but is that a sure thing? Maybe the simulation with e.g. 38 km has the best results in comparison to observations.*

Reply: Interesting question. Given that the time scales of $NO_x$ fates (sources, transport and loss) in the atmosphere are in the order of 1-20 hours, corresponding to a horizontal spatial scale of 4-100 km at ground level considering typical wind speed, prior studies proposed that accurately simulating $NO_x$ requires grid size of ~10 km and for some cases even less (e.g., Valin et al., 10.5194/acp-11-11647-2011, 2011), to realistically capture these processes with such fine temporal and spatial scale.

Since the real atmosphere is horizontally and vertically continuous; in a Eulerian model, higher resolution (e.g., down to the spatial scale of relevant processes) is theoretically more ideal. That being said, if coarser resolutions (e.g., 100 km) somehow

better reproduce observed $NO_x$ while higher resolution simulations (e.g., 10 km) has certain biases (due to the systematic model dependence on resolution as discussed in this paper), it does not mean that the coarse model just works better. Rather, some key processes of $NO_x$ sources and sinks in the coarse model should have compensated for each other to "give the best results".

Line 313-314: "For example, $NO_x$ sources and sinks constrained by matching inadequately coarse simulations with observations could be biased to compensate for the intrinsic model errors discussed here."

We also note several places in the manuscript below, that emphasize the small spatial scales of $NO_x$ processes and the necessary model resolution to capture them:

Line 25-27: "$NO_x$ has strongly localized emissions (Miyazaki et al., 2017; Crippa et al., 2018; Beirle et al., 2019) and relatively short lifetimes (Kenagy et al., 2018; Laughner and Cohen, 2019), which determine its strong spatial heterogeneity, with a short e-folding distance of 30 km and less (Heue et al., 2008; Beirle et al., 2011; Valin et al., 2011)."

Line 30-31: "...it poses as a challenge for chemical transport models (CTMs) to accurately represent the relevant production and loss processes at inter-urban scales (order 10 km) due to limited computational resources."

Line 56-58: " This information urges the necessity to apply adequate resolution that captures the spatial scale of $NO_x$ sources and sinks to simulate and interpret $NO_x$-relevant atmospheric chemistry and air quality issues."

Line 286-288: " Therefore, accurately capturing such regime difference and transition from CTM requires not only accurate emission inventories of $NO_x$ and VOC, but also simulations at representative spatial scales (e.g., 10 km or finer) that correctly distribute these emissions (Valin et al., 2011)."

Line 315-316: " Although this study exploited state-of-science capabilities, biases with respect to resolutions finer than 13 km resolution likely exist considering the highly localized $NO_x$ especially in summer..."

*Are there comparisons with measurement data possible? Can you from the comparisons with TROPOMI conclude which simulation is closest to reality?*

Reply: For sanity check, we performed evaluations of the simulated surface $NO_2$ mixing ratio using hourly observations from the US EPA. Not surprisingly, we found stronger representation of measured $NO_2$ by higher resolution modeling. For example, in January 2015, simulated daytime $NO_2$ concentrations at 181 km resolution have a correlation of 0.50 and normalized mean bias of -53%. Both metrics improve with

increasing resolution to reach a correlation of 0.73 and normalized mean bias of -22% at 13 km resolution. This evaluation validates the reasonable spatial distribution of $NO_x$ emissions from the EDGAR inventory we applied, and of the simulated $NO_x$ concentration. Nonetheless, evaluation of model with observations is significantly out of the scope of this study, and is rather distracting to the audience. We decided to respectfully not include such section in the manuscript.

*Overall, I can absolutely recommend this paper for publication in ACP, in the case that there is a more detailed discussion with regard to the explanation what the reasons are, that the spatial model resolutions has such a high influence to the NOx (and HOx and O3) mixing ratios.*

*Specific comments:*

*Throughout the paper:*

*- You always write of concentrations but I think you are referring most time to mixing ratios, please change this.*

Reply: We note in Line 77-78: "The lowest layer is roughly 120 m thick, with mixing ratios of $NO_x$, $HO_x$ and ozone that we refer to as the "surface concentrations" or "surface mixing ratios" interchangeably."

We mention both terms interchangeably at similar frequency in the revision. "Concentration" is a broader concept, while "mixing ratio" is one measure of "concentration". By mentioning "concentration" in the discussion, we emphasize the broad implications of our results, regardless of the used measure of modeled concentration.

*- In the case that you use ppt or ppb please use instead pptv or ppbv*

Reply: Revised throughout the manuscript.

*- Change "Figure" to "Fig" and "Figures" to "Figs"*

Reply: Revised throughout the manuscript.

*Main text:*

*Page 1, line 1: "The lifetime and concentration of nitrogen oxides (NOx) are susceptible to non-linear production and loss, and consequently to the resolution of a chemical transport model (CTM)." I am not sure if you can really say here "consequently".*

Reply: We deleted "consequently" in the revision (Line 2).

*Page 1, line 2: "Here we use"->"In this study we use"*

Reply: Revised (Line 3).

*Page 1, line 3: "resolutions"->"spatial model resolutions" and "(13-181 km)"->"(six different horizontal grids from 13 to 181 km)"*

Reply: Revised (Line 4-5).

*Page 1, line 4 to 5: Please mention that the GL region is a NOx-saturated area and the SS is a NOx- limited area. Not every reader know this at this point of the paper.*

Reply: We added in Line 8: "...(i.e., $NO_x$-saturated in the GL and $NO_x$-limited in the SS)."

*Page 1, line 5: I would write here "differences to highest resolution of 13 km" instead of "biases" and "-18% to 9%"->"-18% (in GL region) to 9% (in SS region)"*

Reply: Revised as in Line 6-7: "...yielding regional differences (181 km vs. 13 km) of -16% (in the GL) to 7% (in the SS)". The 13 km being the highest resolution has already be implicitly mentioned in Line 4-5 ("six different horizontal grids from 13 to 181 km").

*Page 1, line 14 to 15: As I already said, it would be good also to mention the reason for this model behaviour already in the abstract.*

Reply: We added in Line 1-3: "The lifetime and concentration of nitrogen oxides ($NO_x$) are susceptible to non-linear production and loss, and to the resolution of a chemical transport model (CTM), due to the strong spatial gradients of $NO_x$ and the dependence of its own chemical loss on such gradients."

*Page 1, line 17 and 18: "≡ "->"="*

Reply: We follow the convention in the atmospheric chemistry community to keep using the original symbol, to represent this is a definition.

*Page 2, line 29: "outstanding"->"unexplained"*

Reply: As this topic has already been partially addressed in previous literature, we do not think "unexplained" is suitable to use here. We use "major" instead (Line 32).

*Page 2, line 33: "modeled"->"simulated"*

Reply: Revised (Line 35).

*Page 2, line 40: I would explain Figure S1 here a little bit more and move the figure from the Supplement to the main text.*

Reply: We have moved Figure S1 into Figure A1a in the Appendix.

As in the Introduction Section, we attempt to avoid too detailed mechanistic discussion here. We have introduced Figure A1a in the Result Section comprehensively:

Line 144-147: "...$NO_x$ sources in the GL tend to locate in the $NO_x$-saturated regime (Fig. A1a, right) where concentrated $NO_x$ levels at higher resolutions consume more OH and increase $\tau$; Meanwhile over the SS, the relatively lower $NO_x$ together with the enhanced VOC can reversely promote $HO_x$ production at higher $NO_x$ levels ($NO_x$-limited regime, Fig. A1a, left), thus higher resolution introduces higher OH and lower $\tau$."

Theoretical background about Figure A1a has been well covered and discussed in detail in previous publications (e.g., Page 2 and Figure 1 of Valin et al., 2011). We use

this figure mainly for the interpretation purpose, for which we believe the way illustrated in the manuscript is adequate.

> *Page 2, line 40 and line 50: "resolution"->"spatial resolution"*

Reply: Revised (Line 44 and Line 54).

> *Page 2, line 50: Do you also use different time steps using the different spatial resolutions? If so, I think that could also have an impact on the chemistry and the NOx results.*

Reply: We now clarify the time step in Line 100-102:

"In this work, we apply chemical operator durations of 20 minutes and transport operator durations of 10 minutes (i.e., C20T10) (Philip et al., 2016) to all the simulations, largely in accordance with the GCHP default for cubed-sphere simulations while avoiding interferences to our interpretation from inconsistent operator duration settings."

> *Page 2, line 54: Is the "adequate resolution" always the highest resolution?*

Reply: We added "that captures the spatial scale of $NO_x$ sources and sinks" following "adequate resolution" (Line 57) to be more rigorous.

> *Page 4, line 104: "from space". From which satellite instruments?*

Reply: Different satellite instruments (e.g., GOME, OMI and TROPOMI) have been used to visualize and quantify $NO_x$ sources, therefore "from space" is used here to generalize their joint insights.

> *Page 4, line 108 to 115: You use here the NO2 satellite observations from TROPOMI of the year 2019. Why do you choose the year 2015 for your simulations and not the year 2019?*

Reply: Bottom-up emission inventory is usually not up-to-date. In this study, we use the EDGAR inventory that has emission estimates until 2015. We would like to use the fine-resolution information of scattering weights from TROPOMI, which is available since July 2018.

We added Line 122-124 to justify our usage of 2019 TROPOMI data:

"These mean scattering weights are dependent primarily on observing geometry and relative vertical profiles of molecular and aerosol scattering (Palmer et al., 2001; Cooper et al., 2020), which are expected to be similar in the same month among proximal years."

*Page 4, line 120: "concentrations"->"mixing ratios"*

Reply: Revised (Line 129-130).

*Page 4, line 121: "(upper left)"->"(Fig. 1c, upper left)"*

Reply: Revised (Line 130).

*Page 5, line 127: Where is "further downwind"?*

Reply: We revised here to "further downwind of urban $NO_x$ sources" (Line 136) to be more clear.

*Page 5, line 129: "observable"->"visible"*

Reply: Revised (Line 139).

*Page 6, line 163: Here you write "Another potential cause of the weaker", but you don't really explain the first potential cause before (at least in my opinion, except for the caption of Fig. S1).*

Reply: We clarified discussion of the first cause (varying strengths in the morning and afternoon) in Line 163-170:

"The spatial extent of chemical regimes and their effects on the $NO_x$ biases vary during the course of the day, and contribute to relatively weaker sensitivity of simulated $NO_x$ to resolution in the SS. Fig. A3 shows the resolution-dependence of simulated surface $NO_x$ for morning hours. Relative to the overall effects during afternoon (Fig. 1c), the $NO_x$-saturated regime (with negative $NO_x$ biases at coarser resolution) has broader extent (e.g., intruding further into the south) in the morning hours, meanwhile locations with positive biases ($NO_x$-limited regime) are substantially reduced. The magnitudes of $NO_x$ bias (e.g., 181 km vs. 13 km) are also enhanced over the GL in the morning (-27.8%). These differences are consistent with the relative diel evolution of $NO_x$ (decreases since sunrise) and $HO_x$ (accumulates and peaks after noon) abundances and the consequence on the dominant $HO_x$ loss pathway (e.g., Ren et al., 2003; Ma et al., 2022)."

*Page 6, line 184 and line 187: "faster titration"->"faster O3 titration"*

Reply: We revised the first "titration" term (Line 192). For the second "titration" (Line 196), we referred to both $NO_x$ and $O_3$; We therefore respectively retain the original expression.

*Page 7, line 209 and line 216: "diel"->"diurnal"*

Reply: We believe "diel" represents intraday variability, while "diurnal" indicates day vs. night changes. We therefore respectively retain the term "diel" to emphasize the new insights throughout the 24-hr period from this study.

*Page 8, line 250-251 and line 257 and line 262: You compare here one figures from the main part and one figure from the supplement. I would really transfer Fig. S8 to the main part.*

Reply: We moved Fig. S8 to the Appendix as Fig. A5.

*Page 9, line 278: Means that you should use the highest possible model resolution?*

Reply: The time scales of $NO_x$ fates (sources, transport and loss) in the atmosphere are in the order of 1-20 hours, corresponding to a horizontal spatial scale of 4-100 km at ground level considering typical wind speed, prior studies proposed that accurately simulating $NO_x$ requires grid size of ~10 km and for some cases even less (e.g., Valin et al., 10.5194/acp-11-11647-2011, 2011), to realistically capture these processes with such fine temporal and spatial scale.

Line 286-288: " Therefore, accurately capturing such regime difference and transition from CTM requires not only accurate emission inventories of $NO_x$ and VOC, but also simulations at representative spatial scales (e.g., 10 km or finer) that correctly distribute these emissions (Valin et al., 2011)."

*Page 10, line 300: Here I would add a paragraph explaining the reasons for the different results of the simulations with different resolutions.*

Reply: In the revised manuscript, we have added succinct summary of variable mechanisms driving the resolution dependences.

Line 274-276 (to explain the summer and daytime biases): "At daytime with strong photochemistry, higher resolution modeling more realistically concentrates $NO_x$ emissions near sources, thus decreasing $\tau$ in the $NO_x$-limited regime and increasing $\tau$ in the $NO_x$-saturated regime (Fig. A1)."

Line 289-290 (to explain the winter and nighttime biases): "We found systematic resolution effects of nighttime $NO$-$O_3$ titration efficiency that can drive the $NO_x$ biases over winter (Fig. 2 and Section 3.2), as the anti-correlation between NO and $O_3$ implies faster reaction rates at coarser resolutions."

Line 297-299 (to explain the vertically variable biases): "Our detailed simulation of resolution effects at different altitudes (Figs. 4 and 5) revealed vertically variable sensitivity of $NO_x$ to its chemical loss at different spatial resolutions. These findings significantly enriched the understanding of resolution dependency of satellite columnar observations, in contrast to previous studies that neglected vertical layering."

An additional paragraph that comprehensively summarize these very different mechanisms is hard to generalize. We believe the revised Discussion Section adequately serves the purpose of summarizing these key messages.

*Figures: Fig.1: - Not in every panel of Fig 1c are the green and magenta boxes are drawn. Please change this*

*- Please use ppbv and pptv instead of ppb and ppt*

*Fig.2: - Also here the green and magenta boxes are not drawn in every panel.*

*Fig. S1: - "concentrations"->"mixing ratios"*

*- "ppt" -> "pptv"; "ppb" -> "ppbv"*

*Fig. S3: - "ppt" ->"pptv"; "ppb" ->"ppbv"*

Reply: All these comments are addressed following the suggestions in the revised figures.

*Reviewer 2:*

*The study by Li et al. evaluates the dependency of NOx-chemistry on spatial resolution at different regions with contrast chemical regime, time (daytime vs nighttime, winter vs summer), and vertical layers. The authors have made use of the new capability of a state-of-art chemical transport model GEOS-Chem to conduct the simulations at different resolution. The different regional and temporal NOx-resolution dependences are well explained by the NOx-HOx-ozone chemistry, and the implications for satellite application has been discussed. Overall this is a novel, comprehensive, and nicely-designed study, and is clearly-written. The figures and analyses are high quality. The results have important scientific implications, not only for application of satellite observations to infer NOx concentration and emissions, but also for modelers to understand model bias of NOx and ozone. I recommend publication in ACP after minor revision.*

Reply: Thanks for the positive comments to our work.

*The overall mechanisms to explain NOx-resolution dependency are convincing, but I wonder to whether other factors, such as the difference in meteorological fields at different resolution, may contribute to the NOx-resolution dependency. I understand this might be hard to quantitatively explore but some discussions are beneficial.*

Reply: Non-emission and non-chemistry factors such as transport might exist, but they are fully coupled with the chemistry-driven feedbacks we interpreted in this paper,

thus are hard to disentangle. We added in Line 307-310 of the Discussion Section a clarification of this issue:

"We attribute these systematic simulation biases mainly to the strong localization of $NO_x$ emission and chemistry at a spatial scale of ~10 km and less. Additional modulations from other factors across resolutions, such as sub-grid meteorology-relevant processes (e.g. transport) are also possible, but are fully coupled with the feedbacks revealed in this paper and are non-trivial to disentangle."

We also note in Line 71-75 existing discussion about other factors (including meteorology) besides emissions and chemistry:

"...this capability as demonstrated by Bindle et al. (2021) will not significantly alter our interpretations focusing on discussing redistribution of $NO_x$ emissions and chemical feedbacks, rather than effects from meteorology. Yan et al. (2016) showed that sub-coarse-grid emission-chemical variability dominantly contributed to the differences of simulated tropospheric chemistry between resolutions, overwhelming the effects from resolution of non-chemical factors such as meteorological data."

*Some technical issues*

*(1) Line 79: Are the EDGAR emissions in line with NEI? Why not use EDGAR inventory for VOCs as well?*

Reply: Across the continuous US (CONUS) in 2015, EDGAR $NO_x$ is ~30% higher than the NEI inventory from US EPA, which is within the expected uncertainty of bottom-up emission estimates that depend on the employed activity data and emissions factors. For example, McDonald et al. (10.1021/acs.est.8b00778, 2018) suggested that all mobile sources of $NO_x$ from NEI is 28% higher than the independent FIVE inventory in 2013. Our evaluation of simulated ground $NO_2$ vs. EPA measurements (e.g., correlation of 0.73 and normalized mean bias of -22% at 13 km resolution for January 2015) also confirmed the suitability of EDGAR $NO_x$ emissions inventory used in the simulations.

To our knowledge, EDGAR currently does not have speciated VOC inventory in 2015 for use in a chemical transport model, although total VOC is available. We compared and confirmed that in 2015, total VOC emissions across the CONUS are within 5% among EDGAR, NEI and CEDS.

*(2) Line 135: Would it be helpful to plot the spatial distribution of chemical lifetime of NOx and the change with resolutions?*

Reply: The current version of GCHP does not output chemical removal rates of $NO_x$ of all vertical layers. Frequent $NO_x$ transfer between neighboring pixels via horizontal and vertical transport further complicates the lifetime calculation for each pixel from the model. Alternatively, we included the maps of simulation $HO_x$ (determining the primary chemical $NO_x$ loss at daytime) and its bias at surface level in Figure A2 to aid the interpretation. The opposite responses of $NO_x$ at the two regions and the uniformly positive biases of $HO_x$ demonstrate the validity of our interpretation.

Line 157-158: "Figs. 1c and A2b show that at 13 km resolution, locations with strong $NO_x$ enhancements coincide with locally lower $HO_x$ in the GL, while generally associate with enhanced $HO_x$ in the SS. "

Line 161-162: " Fig. A2b further identifies the broad uniformly positive simulation biases of $HO_x$ in response to the opposite changes of $NO_x$ in the two regions, consistently verifying the two chemical regimes."

*(3) Figure 3: I spend some minutes on understanding Fig.3 and finally find out that I misunderstand the dashed lines (in particular for ozone) as absolute values of concentrations for the coarser resolution. I feel that other readers may be confused as well so I would suggest the authors disregard the relative change in the right axis.*

Reply: The right Y-axis includes important information of biases at other resolutions vs. 13 km throughout the course of a day, a core insight in this section. In the revision, we removed the circles from the dashed lines in Figs. 3 and S3 to make them more distinctive from the absolute (purple) mixing ratios, also making the figure less busy.

*(4) I also feel that I have to jump between the main text and supplementary materials during reading. Please consider moving important figures from SI to the main text.*

Reply: We moved several figures (Figs. S1, S2, S3, S6, S8) originally in the supplement but are more elaborately discussed in the main text, into the Appendix A (Figs. A1, A3-A5). In this way, the different emphasis is conserved while the more important figures can be easily found without referring to the supplement.